TBX5 genetic variants and SCD-CAD susceptibility: insights from Chinese Han cohorts

Rui Yukun 1
Zhou Ju 2
Zhen Xiaoyuan 1
Zhang Jianhua 3
Liu Shiquan shiquan.liu@cupl.edu.cn 4
Gao Yuzhen yuzhengao@suda.edu.cn 1
1 Department of Forensic Medicine, Medical College of Soochow University , Suzhou , China
2 Medical College of Soochow University , Suzhou , China
3 Shanghai Key Laboratory of Forensic Medicine, Institute of Forensic Sciences, Ministry of Justice , Shanghai , China
4 Institute of Evidence Law and Forensic Science, China University of Political Science and Law , Beijing , China
Sezgin Efe
Electronic publication date: 2024 Mar 19
Publication date: 2024
Volume: 12
Electronic Location ID: e17139
Received 2024 Jan 22; Accepted 2024 Feb 28
Copyright: ©2024 Rui et al.
Copyright year: 2024
Copyright holder: Rui et al.
License: This is an open access article distributed under the terms of the Creative Commons Attribution License, which permits unrestricted use, distribution, reproduction and adaptation in any medium and for any purpose provided that it is properly attributed. For attribution, the original author(s), title, publication source (PeerJ) and either DOI or URL of the article must be cited.
License URL: https://creativecommons.org/licenses/by/4.0/

Keywords: Sudden cardiac death, TBX5, rs11278315, Indel polymorphism, Genetic susceptibility

Funding: Natural Science Foundation of China 82171874 81772029 Priority Academic Program Development of Jiangsu Higher Education Institutions This study was funded by Natural Science Foundation of China (Nos. 82171874 and 81772029) and Priority Academic Program Development of Jiangsu Higher Education Institutions. The funders had no role in study design, data collection and analysis, decision to publish, or preparation of the manuscript.

==============================
Background

The prevention and prediction of sudden cardiac death (SCD) present persistent challenges, prompting exploration into common genetic variations for potential insights. T-box 5 (TBX5), a critical cardiac transcription factor, plays a pivotal role in cardiovascular development and function. This study systematically examined variants within the 500-bp region downstream of the TBX5 gene, focusing on their potential impact on susceptibility to SCD associated with coronary artery disease (SCD-CAD) in four different Chinese Han populations.

Methods

In a comprehensive case-control analysis, we explored the association between rs11278315 and SCD-CAD susceptibility using a cohort of 553 controls and 201 SCD-CAD cases. Dual luciferase reporter assays and genotype-phenotype correlation studies using human cardiac tissue samples as well as integrated in silicon analysis were applied to explore the underlining mechanism.

Result

Binary logistic regression results underscored a significantly reduced risk of SCD-CAD in individuals harboring the deletion allele (odds ratio = 0.70, 95% CI [0.55–0.88], p = 0.0019). Consistent with the lower transcriptional activity of the deletion allele observed in dual luciferase reporter assays, genotype-phenotype correlation studies on human cardiac tissue samples affirmed lower expression levels associated with the deletion allele at both mRNA and protein levels. Furthermore, our investigation revealed intriguing insights into the role of rs11278315 in TBX5 alternative splicing, which may contribute to alterations in its ultimate functional effects, as suggested by sQTL analysis. Gene ontology analysis and functional annotation further underscored the potential involvement of TBX5 in alternative splicing and cardiac-related transcriptional regulation.

Conclusions

In summary, our current dataset points to a plausible correlation between rs11278315 and susceptibility to SCD-CAD, emphasizing the potential of rs11278315 as a genetic risk marker for aiding in molecular diagnosis and risk stratification of SCD-CAD.

Introduction

Sudden cardiac death (SCD) refers to the unnatural death attributable to multiple heart disease or non-lethal cardiac impairments, occurring in individuals devoid of any fatal conditions prior to death, typically manifesting within one hour from onset of symptoms (Sampson & Hammers, 2016; Raju et al., 2019). Owing to its unanticipated and irreversible characteristics, SCD presents a formidable global public health challenge with an annual incidence rate of 106.0 per 100,000 individuals in USA, and 40.7 per 100,000 in China (Feng et al., 2018; Tsao et al., 2023). Research on specific mechanisms have been advancing over time, it is generally accepted that SCD is a multifactorial disease, characterized by the interaction of transient triggers and vulnerable substrates, while coronary heart disease serves as a more prevalent substrate (Myerburg, 1993; Deo & Albert, 2012; Semsarian, Ingles & Wilde, 2015). According to survey data, irrespective of national rates, SCD arising from coronary artery disease (SCD-CAD) emerges as the most common cardiac pathology of victims, notably among adults over age 35, while inherited cardiomyopathy and arrhythmic disorders constitute a noteworthy proportion within the pediatric and young adult population (Hayashi, Shimizu & Albert, 2015). In addition, post-mortem genetic testing and molecular investigation have revealed a degree of familial clustering in SCD. Identification of specific single-nucleotide polymorphism (SNP) mutations, coupled with epigenetic investigations, has prompted consideration of the impact of genetic factors in SCD-CAD (Behr et al., 2008; Damani & Topol, 2011; Nurnberg et al., 2016). However, less penetrant phenotypic carriers remained asymptomatic, meaning death is their first and only clinical manifestation with a lack of detailed clinical history and obvious autopsy results upon the cause of death (Sampson & Hammers, 2016; Miles & Behr, 2016). Therefore, to facilitate prospective diagnosis and risk stratification for both affected individuals and the broader population, identification of common genetic risk marker appears indispensable.

T-box 5 (TBX5), a member of the T-box family, is an essential cardiac transcription factor (TFs) and mutations in TBX5 locus are linked to abnormal cardiac function and malformation. For instance, TBX5 missense mutant induces profound transcriptional deregulation and atrial dysfunction (van Ouwerkerk et al., 2022). As one of the first identified monogenic etiologies of familial congenital heart disease, the TBX5 dominant mutation has been reported in Holt-Oram syndrome, known for septation and forelimb defects. Despite with post defects repaired, diastolic ventricular dysfunction still occurs to these patients (Warnes, 2005; Gonzalez-Teran et al., 2022). Also, TBX5 has been found to affect calcium cycling and cardiac diastolic function by directly regulating SERCA2 protein without cardiac structural abnormalities (Zhu et al., 2008). Genome wide association studies (GWAS) elucidates the basic role of TBX5 in arrhythmia and cardiac conduction. In adult mice, TBX5 deletion accounts for diminished expression of cardiac Na channels and connexin 40, leading to severe conduction lagging, ventricular arrhythmias and even SCD (Arnolds et al., 2012; Baruteau, Probst & Abriel, 2015). These findings suggest an extra function of TBX5 beyond embryonic stage. Moreover, recent studies have shown that TBX5 expression has the potential to influence the paracrine microenvironment, reactivate endogenous repair in epicardium, and also exert an impact on inflammation and epicardial adipose tissue associated with atherosclerosis (Ruiz-Villalba & Pérez-Pomares, 2012; Quijada, Trembley & Small, 2020). Overall, TBX5 is of great importance in cardiac pathology and physiology, but little is known concerning SCD-CAD predisposition in terms of genetic mechanisms.

In this study, based on a systematic filtering strategy, an eight-bp insertion/deletion (indel) polymorphism in the 500-bp downstream region of TBX5 was screened to investigate its correlation with SCD-CAD susceptibility in Chinese populations from four different geographical regions of China. Further functional experiments exploring the underlying mechanisms were performed.

Materials & Methods

Ethics statement

Ethical approval for this study was granted by the Ethical Committee of Soochow University (approval number: ME81772029). Prior to the investigation, written informed consent was acquired from all healthy blood donors and the victim’s family.

Study populations

Our study included a cohort of 553 healthy controls and 201 SCD-CAD cases, all of whom were genetically unrelated Han Chinese from central-southern, eastern, southern and southwestern China, respectively. The SCD-CAD case set was gathered from 2012 to 2020 at Soochow University, Institute of Forensic Science at Guizhou Medical University, Shanghai Key Laboratory of Forensic Medicine at Ministry of Justice, the Medicolegal Expertise Center of Sun Yat-sen University and the Forensic Medical Identification Center of Central South University. The criteria for both sample inclusion and exclusion align with those previously described (Wang et al., 2017; Zou et al., 2019). Rigorous forensic pathological investigations were undertaken to rule out any fatal features except for coronary atherosclerosis of varying degrees, thereby attributing the sudden deaths to coronary heart diseases. Comprehensive toxicological examinations were also conducted to avoid the influence of poisoning and pharmaceutical substances. Age (±5 years) and gender matched 553 healthy controls were enrolled through community nutrition information questionnaires performed during the same time frame within identical areas. Venous blood was taken simultaneously from those healthy controls after written informed consent. These healthy donors had no family history of cardiovascular disease or sudden death. A collection of 23 cardiac tissues from healthy human donors was gathered at the Medicolegal Expertise Center of Soochow University and Guizhou Medical University during traffic accident autopsy. Upon retrieval through medicolegal autopsy, obtained tissues were promptly preserved at −80 °C for following protein, DNA or RNA extraction.

Bioinformatic analysis

Variants within TBX5 gene were sourced from dbSNP database (Sherry et al., 2001). Putative functionality of these variants were examined using RegulomeDB and HaploReg databases (Ward & Kellis, 2012; Boyle et al., 2012). Analysis of splicing quality trait loci (sQTL) was conducted utilizing data from the Genotype-Tissue Expression (GTEx) database (GTEx Consortium, 2013). Linkage disequilibrium (LD) analysis was performed applying R package (version 4.0.3; R Core Team, 2020), and genotype data was supplied by Ensembl Genome Browser (Chan, 2018; Martin et al., 2023). JASPAR was employed for transcription factors binding site (TFBS) prediction within the LD block, and the results were visualized by UCSC genome browser (Castro-Mondragon et al., 2022). Binding sites predicted in the JASPAR CORE collection with scores above 400, indicative of a p-value less than 0.0001, were included. GRNdb and ChIPBase databases were utilized to predict potential target genes of TBX5 (Zhou et al., 2017; Fang et al., 2021). Gene ontology (GO) annotations were performed using the DAVID database (Sherman et al., 2022). Additionally, the STRING database (Version 11.5) was employed to predict protein-protein interaction (PPI) network (Szklarczyk et al., 2023).

DNA extraction and genotyping

Genomic DNA was extracted from both blood samples and human myocardium tissues employing the Blood DNA Kit and Blood spot DNA kit (TIANGEN). A pair of genotyping primers (Forward: ACTTTTCTTCTCCAGTGCCTAC, Reverse: CCATTGTCCTTGAGTGTGAT) designed to amplify the polymorphic region containing rs11278315 were synthesized by Genewiz Company (Suzhou, China). For detailed primer information, please refer to Table S1. The products of the polymerase chain reaction were determined by 7% nondenaturing polyacrylamide gel electrophoresis (PAGE) and observed through silver staining method, during which a double-blinded approach was adopted (Allen, Graves & Budowle, 1989). Regarding quality control, a random selection of 50 samples were subjected to direct sequencing and randomly selected 10% of all DNA samples were verified independently by two investigators.

RNA extraction and quantitative real-time PCR (qRT-PCR) analysis

Total RNA from human myocardial tissues, extracted by TRIzol reagent (Invitrogen), was subsequently quantified with the NanoDrop (Thermo Scientific). The A260/A280 ratio was 1.8−2.3. According to NCBI database, we identified three experimentally validated isoforms of TBX5: NM_000192.3 (variant 1), NM_080717.4 (variant 3), and NM_181486.4 (variant 4). Variant 1 as the longest variant encodes the same protein as variant 4, whereas variant 3 lacks an exon containing the transcription start site, resulting a protein with a shorter N-terminus. Due to the high sequence similarity among three isoforms and the limited suitability of specific fragments for primer design, we finally selected a pair of universal primers capable of amplifying all three isoforms, as well as a pair of primers that specifically amplify variant 4, a representative isoform suggested by Matched Annotation from the NCBI and EMBL-EBI (Morales et al., 2022). The primer sequences designed for isoforms and GAPDH were given below and the specificity of primers were confirmed with Primer-BLAST:

TBX5-Forward (variant 1, 3, 4): TAGATTACACATCGTGAAAGCGG, TBX5-Reverse (variant 1, 3, 4): GGAAAGACGTGAGTGCAGAAC;

TBX5-Forward (variant 4): AGTAAACCCCGCATAAACCCC, TBX5-Reverse (variant 4): GCAAGGTTCTGCTCTCGTTCG;

GAPDH-Forward: CTCTCTGCTCCTCCTGTTCGAC, GAPDH-Reverse: TGAGCGATGTGGCTCGGCT.

Quantitative real-time PCR (qRT-PCR) analysis

Following reverse transcription with the HiScrip III All-in-one RT SuperMix Perfect for qRT-PCR (Vazyme), acquired cDNA was used immediately for ChamQ Universal SYBR qRT-PCR Master Mix (Vazyme) employing the LightCycler96 instrument (Roche) with a minimum of twice replicates. Remaining RNA and cDNA were stored at −80 °C. Reaction mixes with a total volume of 20ul were prepared from 10 µl ChamQ Universal SYBR qRT-PCR Master Mix, 7.84 µl RNase-free water, 0.08 µl forward primer (50µM), 0.08 µl reverse primer (50µM) and 2 µl cDNA per well. The thermal cycling conditions was as specified by the manufacturer: initial denaturation at 95 °C for 30 s, followed by 40 cycles of 95 °C for 10 s and 60 °C for 30 s. Melting curve analysis was performed as follow: 95 °C for 15 s, 60 °C for 60 s and 95 °C for 15 s. Standard curve was generated to test amplification efficiency of over 90%. Set GAPDH as internal control and samples with ins/ins plus ins/del genotype as control group for the 2−ΔΔCT algorithm-based calculation of quantitative results, where Cq refers to cycle threshold. The formulas used was as follow: 2−ΔΔCT = 2−[(Cq of target gene−Cq of GAPDH)experiment−(Cq of target gene−Cq of GAPDH)control]

Western blot assay

Total protein in human myocardium tissues were extracted using Phenylmethanesulfonyl fluoride and Radio Immunoprecipitation Assay. Homogenized protein(60ug) was separated via 10% SDS-polyacrylamide gel electrophoresis (SDS-PAGE) and subsequently transferred onto a PVDF membrane (Millipore). After a two hours block at room temperature and overnight incubation with the primary antibodies Anti-TBX5 (Rabbit, Cat# 13178-1-AP, 1:500, Proteintech) and Anti-GAPDH (Mouse, Cat# AB-M-M001, 1:1000, Goodhere Biotechnology) at four degrees centigrade, the probed proteins were then treated with the secondary antibody (1:2000, Biyotime Biotechnology) for one hour. The hypersensitive Enhanced Chemiluminescence (ECL) device was used to visualize bands. The Western blot results were finally analyzed using ImageJ analysis software (National Institutes of Health).

Cell cultures

Human embryonic kidney 293T (HEK 293T) cell lines were cultured in Dulbecco’s Modified Eagle Medium (Procell) supplemented with 10% fetal bovine serum (Inner Mongolia Opcel Biotechnology Co., Ltd) and 1% penicillin-streptomycin (Invitrogen) at 37 °C in a humidified incubator with 5% CO2. The cell lines were originated from Shanghai Cell Bank of Chinese Academy of Sciences. The cell lines were characterized by Genetic Testing Biotechnology Corporation (Suzhou, China) using short tandem repeat (STR) markers.

Reporter plasmid vector construction and dual Luciferase reporter assay

DNA segments containing 326 or 318 base pairs, centered around rs11278315, were directly synthesized and subcloned into XbaI and FseI sites of pGL3-control vector by Genewiz Company (Suzhou, China). This process yielded vectors containing wild-type (pGL3-WT) with an insertion allele or mutant-type (pGL3-MT) with a deletion allele, with orientation and sequence authenticated by direct sequencing. After 24 h of cell seeding in 24-well plate (1 × 105 cells per well), a co-transfection by the jetPRIME transfection reagent (Polyplustransfection) was completed by adding 400 ng of the reconstructed pGL3-WT or pGL3-MT vector along with 20 ng of the pRL-SV40 vector (Promega). Empty pGL3-control vector was also transfected as a negative control. Cell lysis was performed at 24 h post transfection, followed by disruption in 100 µL of Passive Lysis Buffer (Promega). Measurement of Firefly and Renilla luciferase activity was performed using SpectraMax iD3 (China), with a minimum of eight wells per conditions measured.

Statistical analysis

IBM SPSS Statistics Subscription (version 27.0.1) and GraphPad Prism (version 9.0.0) were carried out for statistical analysis. Hardy-Weinberg equilibrium assessed by chi-square test was utilized to ensure the representativeness of genotype distribution in control samples. The odds ratio (OR) and 95% confidence interval (95% CI) were calculated through unconditional logistic regression, with age and gender adjusted, so as to look into the relationships between rs11278315 and the risk of SCD-CAD. Student’s t-test was used for comparing the means of two groups for functional evaluation. In our study, we set up p < 0.05 as statistical significance threshold and all statistical tests were two-sided.

Results

TBX5 gene downstream candidate variants selection

Figure 1 depicts the flowchart employed in our current study. Given the potential regulatory role and the presence of various functional elements of downstream non-coding regions in gene expression, we conducted a screening of the 3′ UTR and 500-bp downstream region of TBX5. In Table S2, we have listed all 11 variants with a minor allele frequency (MAF) > 0.05. Based on the compatibility with the widely used capillary electrophoresis platforms in forensic genetics, two selected indel sites, rs35534655 and rs11278315, which exhibited length polymorphism characteristics analogous to STR markers, were picked as candidate variants. Nonetheless, rs35534655 is a multiple adenine (A) duplicate that make genotyping complicated. Therefore, we chose rs11278315 as our candidate variant. Through LD analysis within 10 kb of rs11278315, we discovered that rs11278315 shared a LD block with six other sites (Fig. 2A), which were listed in Table S3 . Among these sites, rs11278135 is strongly associated with rs1895597 and rs3825214, which was associated with electrocardiographic traits and atrial fibrillation in GWAS analysis.

Figure 1 The workflow employed in our current study.

500BP, 500 base-pairs downstream region; MAF, minor allele frequency; sQTL, splicing quality trait loci; GTEx, Genotype-Tissue expression; RegulomeDB and HaploReg, two databases for functional annotation; SCD-CAD: sudden cardiac death originated from coronary artery disease.

Figure 2 LD analysis and histone modification markers of rs11278315.

(A) Linkage disequilibrium (LD) analysis of rs11278315 and its nearby loci, including adjacent Genome wide association study (GWAS) sites. (B) The histone modification markers of rs11278315 Haploreg database. Yellow line represents polymorphic site rs11278315.

Database retrieval and functional prediction analysis

Database annotation suggested that rs11278315 was enriched with H3K36me3 (associated with active gene transcription) or H3K27me3 (Inhibitory promoter related), H3K4me1 (only found in enhancers) (Fig. 2B). Besides, we predicted 17 TFBS (Table S2) within the LD block of rs11278315 by using JASPAR database. Then we performed the GO enrichment analysis on these predicted sites by querying DAVID database. Significance was defined as p < 0.05, and the most enriched GO terms are presented as the -log10 of the uncorrected p-values. GO analysis was divided into three aspects: biological process (BP), and cellular component (CC) and molecular function (MF). In terms of BP level, the top terms primarily pertained to regulation of transcription from polymerase II promoter and regulation of gene expression, whereas the most enriched terms consist of chromatin, nucleus, and transcription factor complex in CC. At the MF level, the enriched terms included transcription factor, activator activity, and RNA polymerase II regulatory region sequence-specific DNA binding (Fig. 3A).

Figure 3 Function prediction analysis.

(A) Gene ontology (GO) analysis of 17 TFBS within LD block predicted by JASPAR, with Homo sapiens as the default species, p < 0.05. (B) Venn Diagram of 43 intersecting target genes predicted by GRNdb and ChIPBase databases. (C) GO analysis of 43 intersecting target genes, with Homo sapiens as the default species, p < 0.05. (D) protein-protein interaction (PPI) network diagram of cooperative TF interaction involving TBX5, confidence score ≥ 0.7, default tissue is human heart. Network nodes represent proteins and edges represent protein–protein associations.

In addition, given that TBX5 is a transcription factor, we predicted 43 downstream target genes from databases intersection (Fig. 3B) and observed following degrees of enrichment: cardiac conduction system development, regulation of cardiac muscle contraction, ventricular cardiac muscle tissue morphogenesis at the BP level; sarcomere and neuromuscular junction at the CC level; epidermal growth factor receptor binding, protein serine/threonine/tyrosine kinase activity, ATP binding at MF level (Fig. 3C). In the context of protein expression, we analyzed the functional and physical protein associations related to TBX5, and generated a set of protein-protein interaction (PPI) network based on the STRING database with a high confidence score of ≥ 0.7 (Fig. 3D).

The associations of rs11278315 with SCD-CAD susceptibility

Table 1 summarized the demographic characteristics of individuals who participated in current investigation. The detailed features and forensic autopsy findings regarding the SCD-CAD cases were listed in Table S4. Of note, cases exhibited a higher prevalence in males, having a 9.05:1 male to female ratio. Among all cases, 44 cases (24.31%) occurred following strenuous exercise or significant physical exertion, while 60 cases (33.15%) took place in stress-inducing conditions such as emotional events, arguments, alcohol consumption, injury, and surgical treatments, categorizing them as stress-related stimuli. Additionally, 16 cases (8.84%) transpired during sleep, and 81 cases (44.75%) occurred in a tranquil state or lacked eyewitnesses, which were classified as nonspecific.

Table 1 Clinical characteristics of SCD-CAD cases and controls.

Characteristic	SCD-CAD	SCD matched controls	
No. of individuals	201	553	
Sex, No.			
Male	181	477	
Female	20	76	
Age, mean ± SD (range)			
Overall	51.92 ± 13.89 (18–92)	47.28 ± 14.23 (15–90)	
Males	51.64 ± 13.14 (18–87)	46.03 ± 13.57 (15–90)	
Females	60.00 ± 17.74 (27–92)	55.08 ± 15.81 (24–87)	
Events at sudden death (SD)			
Nonspecific	81		
Physical activity	44		
Stress	60		
Sleep	16		
Symptoms before SD			
None	124		
Others	77		
Megalothymus			
Positive	4		
Negative	197		

As illustrated in Fig. 4, the genotyping and sequencing results for rs11278315 confirmed its polymorphic character. The observed rs11278315 genotype frequencies in control group were consistent with Hardy-Weinberg equilibrium. Allele and genotype frequencies of rs11278315, and odds ratio (OR) with its 95% confidence interval (CI) for both groups, were listed in Table 2. According to codominant model, the del/del genotype was linked with a significantly reduced risk of SCD-CAD when compared to individuals with the ins/ins homozygous allele (OR: 0.47, 95% CI [0.29–0.75], p = 0.0014), in line with recessive and additive models (OR: 0.64, 95% CI [0.44–0.93], p = 0.0194; adjusted OR: 0.70, 95% CI [0.55–0.88] p = 0.0019). As described, these findings indicated a significant association between rs11278315 and susceptibility to SCD-CAD, with the deletion allele lowering the risk of SCD-CAD in its carriers.

Figure 4 Example output from sequencing and genotyping assays of rs11278315.

(A) The sequencing results of insertion, deletion allele in template strands. The underlined bases indicate the “GTGGATCA” insertion in coding strands. (B) The genotyping outcomes by using 7% non-denaturing polyacrylamide gel electrophoresis (PAGE) and silver staining (lane 4, 8 and 9, ins/ins genotype; lane 2, 3, 5, 10 and 12, ins/del genotype; lane 1, 6, 7 and 11, del/del genotype; lane *, DNA Marker).

Table 2 Associations between rs11278315 and sudden cardiac death susceptibility in case control sets recruited during 2012–2020.

Genetic Model	Genotype	Cases	(%)	Control	(%)	OR (95% CI)a	P value	
Codominant model	ins/ins	54	26.87%	97	17.54%	1.00 (Reference)		
	ins/del	103	51.24%	287	51.90%	0.65 (0.43–0.96)	0.031579	
	del/del	44	21.89%	169	30.56%	0.47(0.29–0.75)	0.001370	
	P trend						0.001541	
Recessive model	ins/ins+ins/del	157	78.11%	384	69.44%	1.00 (Reference)		
	del/del	44	21.89%	169	30.56%	0.64 (0.44–0.93)	0.019378	
Additive model	ins allele	211	52.49%	481	43.49%	1.00 (Reference)		
	del allele	191	47.51%	625	56.51%	0.70 (0.55–0.88)	0.001933	
Notes.

CI confidence interval

OR odds ratio

a Adjusted for age and gender factors.

The correlation between rs11278315 genotype and expression of TBX5

Based on GTEx database, we identified these variants rs11278315, rs7977083, rs10744823, rs12367410, rs3825214 within the LD block as sQTL variants of TBX5 (Fig. 5). After genotyping cardiac tissues from healthy human donors, we identified 11 tissue samples with del/del, four with ins/ins and eight with ins/del genotype. Our qRT-PCR analysis using these 24 same normal myocardial samples showed that individuals carrying deletion allele had significant lower expression level compared to those carrying insertion allele (Fig. 6A). Intriguingly, variant 4, when carried with the deletion allele, exhibited relatively lower expression level compared to results from total isoforms. As the results from total isoforms showed a 0.73-fold lower expression (p = 0.0002), while results from variant 4 showed a 0.41-fold lower expression (p < 0.0001), with the ins/ins and ins/del genotype set as reference. For another, western blot analysis also correspondingly reflected considerably higher-level trend of TBX5 protein with the ins/ins genotype of rs11278315 compared to the ins/del and del/del genotypes (p >  0.05) (Fig. 6B). Although the results were not statistically significant, this may be due to our small sample size and inner differences between samples. Collectively, those indicates that rs11278315 has a regulatory effect on TBX5 expression.

Figure 5 The sQTL analysis within LD block.

The Splicing quantitative trait loci (sQTL) analysis between TBX5 and rs11278315, rs7977083, rs10744823, rs12367410, rs3825214 based on GTEx database.

Figure 6 The expression levels of TBX5 in human myocardium tissues with different genotypes.

(A) The mRNA level of total isoforms in human myocardium samples showed that in tissues with del/del genotype was a 0.73-fold lower than that in samples with ins/ins and ins/del genotype (*** p = 0.0002), while results from variant 4 showed a 0.41-fold lower expression (****p < 0.0001). del/del, N = 11, ins/ins and ins/del, N = 12. (B) Western blot analysis of TBX5 protein level in human myocardium tissues with different genotypes also indicated a trend that the del/del genotype individuals have lower expression level. Lanes 1–3 represent ins/ins genotype, and lanes 4–6 represent ins/del genotype, and lanes 7–9 represent del/del genotype.

The influence of rs11278315 on gene transcription activity

With respect to the correlation between rs11278315 and gene transcription activity, we performed a dual luciferase reporter assay to assess relative fluorescence intensity at 24 h after transfection. It is noteworthy that either pGL3-WT or pGL3-MT transfected group presented higher luciferase activity than negative pGL3-control vector co-transfected concurrently. Likewise, the group transfected with pGL3-MT displayed a significantly lower luciferase activity in comparison to the group transfected with pGL3-WT, highlighting the regulatory properties of rs11278315 variant (Fig. 7).

Figure 7 The effect of rs11278315 on gene transcriptional activity.

The relative firefly luciferase activities were compared between insertion construct group (pGL3-WT) and deletion construct group (pGL3-MT) in 293T cell lines. Cells transfected with pGL3-MT exhibited a considerably lower luciferase activity as compared with cells transfected with pGL3-WT (*p < 0. 01).

Discussion

TBX5, a pivotal transcription factor involve in cardiac specification and heart morphogenesis in vertebrate, undergoes strictly restricted dynamical spatial and temporal regulation during various stages of heart development. TBX5 haploinsufficiency has been linked to Holt-Oram Syndrome, and part of the patients have been reported to display coronary artery abnormality (Liberatore, Searcy-Schrick & Yutzey, 2000; Steimle & Moskowitz, 2017). There is consensus that epicardium is the origin of annulus fibrosus, adventitial fibroblasts and interstitial, then differentiates into the coronary arterial smooth muscle cells (Quijada, Trembley & Small, 2020). Studies on animal models have revealed that both overexpression and deficiency of TBX5 in the embryonic proepicardial organ (PEO) disrupted proepicardial cell migration and reduced epicardial cell proliferation, so that brought delayed adhesion of epicardial cell to myocardium and recruitment disorders of coronary vascular smooth muscle cell, leading to abnormal coronary vasculogenesis (Gittenberger-de Groot et al., 2012; Diman et al., 2014). Meanwhile, conditional TBX5 deletion induced cardiac dysfunction and arrhythmias led to a high mortality rate due to SCD in mice, while TBX5 expression normalization could have therapeutic value. Further, TBX5 expression from human adult ischemic or dilated cardiomyopathy were substantially abundant, indicating its participation may beyond congenital heart disease (Rathjens et al., 2021). However, our study discovered that deletion allele of rs11278315, linked to lower expression, might confer protection against SCD-CAD. Either way, dysregulation of gene expression, whether upregulated or downregulated, might disrupt heart development and impair maintenance of cardiac function in adulthood and bring malignant effects on cardiovascular system (Hori et al., 2018).

Furthermore, TBX5 has been testified to cooperate with various cardiac transcription factors (TFs). For instance, the strong interaction between GATA4 and TBX5 is closely linked to the proper function of cardiac super-enhancers (Ang et al., 2016). Our PPI network analysis also provided valuable insights into cooperative TFs interaction involving TBX5. Fine-tuned interactions among TFs underpin the foundation of normal cardiac function, while deviations in their expression levels and alterations in their binding domain structures have the potential to precipitate cardiac pathologies (Nemer & Nemer, 2001; Wang et al., 2015). In addition, GO analysis of TBX5 potential target genes in heart tissue uncovered that changes in TBX5 expression may contribute to dysregulation of genes involved in heart contraction, cardiac morphogenesis and conduction system development. Remarkably, the cardiac conduction and arrhythmia correlation of rs11278315 highly linked SNPs has been supported by GWAS analysis. Based on the above, we hypothesize that due to the long-term allele-based dynamic deviation in TBX5 expression levels from development into adulthood, coupled with disruption in relative dose homeostasis among cardiac TFs and their regulatory interactions with downstream target genes, subtle and imperceptible defects or deficiencies were left behind. These factors could give rise to a more fragile coronary artery matrix, ultimately amplifying the risk of SCD-CAD.

Alternative splicing (AS) is a mechanism that generates isoforms containing different exons of same gene. Functional annotations of rs11278315 and entire LD block unveiled comparable enrichment of H3K36me3. H3K36me3 was considered to influence mRNA alternative splicing through the recruitment of effector proteins (dela Mata et al., 2003). This points to the possibility that a mutation in rs11278315 may alter the initial splicing pattern by changes to splicing effector protein recruitment. Correspondingly, the qRT-PCR results indicated that variant 4 carried with the deletion allele exhibited a lower expression level compared to that of total isoforms carrying the deletion allele. That means normal human myocardial tissues with the deletion allele had comparatively higher expression level of variant 1, variant 3 or other undefined isoforms. Specially, variant 1 encodes same protein to variant 4, while variant 3 lacks an exon containing the transcription start site, resulting a protein with a shorter N-terminus. It has been reported that the removal of the N-terminus of TBX5 led to a decrease in synergy with GATA-4 (Georges et al., 2008). In other words, the preservation of protein structure is required for the strength and duration of physical interactions, as well as subsequent function. Therefore, we speculate imbalanced expression of different effector isoforms might cause functional alterations and increase susceptibility to SCD-CAD. This observation may also elucidate why our study showed a protective role of the deletion allele, in contrast to previous findings, as variations in the expression level of certain effector isoforms could also be reflected by changes in overall expression level. In addition, combined with GO analysis of TFBS in the LD block, the enrichment closely correlated with the regulation of RNA polymerase II promoter and sequence-specific DNA binding. Taken the other four sQTL sites mentioned previously together, the entire LD block may collectively engage in the processes of alternative splicing and genes transcriptional regulation.

Our study still has a few limitations. Due to obstacles in assembling samples of human cardiac tissue and the rarity of SCD-CAD incidence, future studies with larger sample scale are needed for proof of our current findings. Considering the differential genetic ancestries in the natural populations, a region-specific stratification analysis is needed upon a sufficient number of SCD-CAD cases from different regions enrollment. Moreover, further functional experiments are required to dissect the functional distinctions among different isoforms and elucidate alternative splicing as well as cardiac transcription regulation mediated by the entire LD block.

Conclusions

To sum up, the findings of present study, for the first time, provided the initial evidence that the novel variations rs11278315 of TBX5 are closely associated with SCD-CAD susceptibility in Chinese populations. Consequently, the indel polymorphism rs11278315 might act as a prospective genetic risk marker for molecular diagnosis and risk stratification of SCD-CAD.

Supplemental Information

Supplemental Information 1 All primer sequences used for DNA genotyping and qRT-PCR

Supplemental Information 2 Variants inormation

All variants with a minor allele frequency (MAF) > 0.05.

Seven sites in a LD block.

Supplemental Information 3 Predicted TFBS within the LD block of rs11278315

Supplemental Information 4 The detailed features and forensic autopsy findings regarding the SCD-CAD cases

Supplemental Information 5 Cell line authentication of HEK293T

Supplemental Information 6 MIQE checklist

Supplemental Information 7 Raw data

We thank all the participants in this study.

Additional Information and Declarations

Competing Interests

Author Contributions

Human Ethics

Data Availability

The authors declare there are no competing interests.

Yukun Rui performed the experiments, prepared figures and/or tables, authored or reviewed drafts of the article, and approved the final draft.

Ju Zhou performed the experiments, prepared figures and/or tables, authored or reviewed drafts of the article, and approved the final draft.

Xiaoyuan Zhen analyzed the data, prepared figures and/or tables, and approved the final draft.

Jianhua Zhang analyzed the data, authored or reviewed drafts of the article, and approved the final draft.

Shiquan Liu conceived and designed the experiments, authored or reviewed drafts of the article, and approved the final draft.

Yuzhen Gao conceived and designed the experiments, prepared figures and/or tables, authored or reviewed drafts of the article, and approved the final draft.

The following information was supplied relating to ethical approvals (i.e., approving body and any reference numbers):

The Soochow University granted Ethical approval to carry out the study within its facilities (Ethical Application Ref: ME81772029).

The following information was supplied regarding data availability:

The basic information of variants and detailed infomation of SCD-CAD cases are available in the Supplementary File.

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
