# Peer review of "TBX5 genetic variants and SCD-CAD susceptibility: insights from Chinese Han cohorts"

_PeerJ, doi:10.7717/peerj.17139_

## Round 0.1 · original submission · Minor Revisions

Dear Authors,

Reviewers suggested revisions for your manuscript. Please address all the points and resubmit your manuscript.

Sincerely,


Reviewer 1 ·

Basic reporting

no comment

Experimental design

- In study populations, you mentioned that 553 healthy controls were enrolled through community nutrition information questionnaires. Could you provide details on the locations where you recruited these healthy controls? Did they come from the same regions of China? How many of the cases and controls were Northern or Southern Han Chinese? Northern and Southern Han Chinese share pretty different genetic ancestries. Could you adjust for this variable in your logistic regression model?
- You mentioned only 23 cardiac tissues were collected from healthy human donors. So you only have DNA information on 23 of 553 healthy donors? if not, please describe how and when the blood samples were collected from the healthy donors.
- You have found 11 variants that have MAF > 0.05 among downstream regions of TBX5. Have you considered conducting logistic regression analyses for all 11 SNPs and adjusting for the p-values for multiple comparisons? I wonder if there will be other variants that stand out and have a more significant association with SCD than rs11278315.
- Were the p-values shown in Figure 5 adjusted for multiple comparisons?

Validity of the findings

no comment

·

Basic reporting

The manuscript maintains clear and professional language throughout, with thorough literature references providing essential context. It follows a well-structured format, with informative figures and tables aiding data interpretation. The study's results directly address hypotheses, focusing on the association between the rs11278315 variant of TBX5 and SCD-CAD susceptibility, supported by robust data and contextualized within existing literature.

Experimental design

The study conducted a comprehensive investigation into the association between TBX5 gene variants and susceptibility to sudden cardiac death due to coronary artery disease (SCD-CAD) in Han Chinese populations. Multiple methodologies were employed, including bioinformatic analysis, quantitative real-time PCR analysis, Western blot assays, cell culture experiments, and dual luciferase reporter assays. The results revealed a significant association between the rs11278315 variant of TBX5 and reduced risk of SCD-CAD, supported by thorough statistical analysis and robust experimental validation. These findings underscore the potential of rs11278315 as a genetic marker for molecular diagnosis and risk assessment of SCD-CAD.

Validity of the findings

While the study does not evaluate impact and novelty, it advocates for meaningful replication, providing robust and statistically sound data, fostering credibility. Conclusions are tightly linked to the original research question, supported by controlled data, thus enhancing the study's validity and relevance within the scientific community.

Additional comments

Overall, the manuscript presents compelling evidence for the association between rs11278315 and SCD-CAD susceptibility, offering valuable insights into the genetic basis of cardiac disorders. Addressing the noted weaknesses through additional validation and replication studies would enhance the robustness and applicability of the findings.

---

## Round 0.2 · accepted · Accept

Reviewers found your manuscript acceptable. Please follow directions from the PeerJ staff for timely publication of your manuscript.